# Optimizing the Methodology for Antioxidant Activity Analysis of Manuka Honey

**DOI:** 10.3390/foods14081341

**Published:** 2025-04-14

**Authors:** Xiaoling Zhou, Wei Zhao, Zihong Ye, Jintian Tang, Yafen Zhang

**Affiliations:** Key Laboratory of Microbial Quantitative Detection and Biological Product Quality and Safety, State Administration for Market Regulation, College of Life Sciences, China Jiliang University, Hangzhou 310018, China; zxl2297355251@163.com (X.Z.); 18755784636@163.com (W.Z.); zhye@cjlu.edu.cn (Z.Y.); jintiantang@cjlu.edu.cn (J.T.)

**Keywords:** Manuka honey, antioxidant activity, methodology optimization, CAA assay, reproducibility

## Abstract

Manuka honey (MH) is increasingly recognized for its potent antioxidant properties, making it a promising functional food ingredient. However, discrepancies in assessment methods have impeded the standardization of its antioxidant capacity. This study compared DPPH, ABTS, and cellular antioxidant activity (CAA) assays under varying conditions to identify the most reliable approach for assessing MH’s antioxidant properties. The results showed that the reaction temperature for the chemical method, setting it at 37 °C, enhanced the antioxidant capacity of MH. For the cellular assay, we optimized honey concentration, injury duration, damage model, and cell model. The result showed that sugar-reduced MH achieved the same high efficiency as the chemical method. A stable cellular assay method was established in HepG2 cells, offering superior reproducibility with an intra-RSD of 4.83% (<5%) and an inter-RSD of 7.51% (<10%). Additionally, studies have found that methyl syringate (MSY), a key polyphenolic compound in Manuka honey (MH), exhibits extremely high antioxidant activity. However, due to its low concentration, its overall contribution to the honey’s antioxidant activity is limited. This optimized CAA-based approach provides reliable technical support for the accurate evaluation of the antioxidant activity of MH.

## 1. Introduction

Manuka honey (MH), derived from *Leptospermum scoparium* of the Myrtaceae family in New Zealand, has long been recognized for its exceptional nutritional value and diverse biological properties [1]. These properties, particularly its antibacterial and anti-inflammatory effects, have been widely studied and are attributed to its unique chemical composition [2,3]. The antibacterial activity of MH is strongly correlated with the concentration of methylglyoxal (MGO), with studies reporting a correlation as high as 98% [4,5]. In addition, the Unique Manuka Factor (UMF) is a widely recognized grading system used to evaluate the quality of MH. It assesses the antibacterial potency by quantifying key bioactive compounds, including MGO and dihydroxyacetone (DHA), which serve as indicators of its purity and therapeutic efficacy [6]. In recent years, increasing attention has been directed toward the antioxidant activity of MH, which plays a critical role in mitigating oxidative stress and preventing the progression of chronic diseases, such as cancer, cardiovascular diseases, and neurodegenerative disorders [7,8]. However, there is currently no consensus on the most appropriate methods and indicators for evaluating antioxidant activity. Although higher MGO content is closely associated with stronger antibacterial activity, its relationship with antioxidant capacity remains unclear. For example, MH with an MGO content of 250 mg/kg showed slightly higher antioxidant activity than MH with 400 mg/kg, suggesting that increased MGO levels do not always correlate with enhanced antioxidant effects [9].

Studies by Alvarez-Suarez have demonstrated that the phenolic compounds in MH exhibit significant antioxidant activity by effectively scavenging free radicals and reducing oxidative damage [10]. These findings emphasize the vital role of polyphenolic compounds in the antioxidant potential of MH. However, Tananaki et al. [11] reported that despite MH having a high total phenolic content, its overall antioxidant capacity was moderate to low compared to seven other monofloral honey types. Similarly, Goslinski et al. [12] found that although MH contained a relatively high level of total polyphenol content, its DPPH radical scavenging activity was lower than expected. In comparison, buckwheat honey, which also has a high polyphenol content, exhibited an unexpectedly low antioxidant index. These inconsistencies suggest that the relationship between total polyphenols and antioxidant capacity is complex and not always correlated. It also implies that polyphenols from fruits as a functional additive to honey because it is unclear in the current version. Additionally, while polyphenols from fruits as a functional additive to honey may enhance its measured antioxidant activity, the correlation between polyphenol content and antioxidant capacity in honey does not exhibit a significant linear relationship [13,14].

Among the antioxidant phenolic compounds present in MH, methyl syringate (MSY) has emerged as a promising component. MSY contains both hydroxyl (-OH) and methoxy (-OCH_3_) groups and has been shown to enhance reactive oxygen species (ROS) scavenging and inhibit oxidative damage by modulating oxidative stress pathways [15,16]. While MSY’s antioxidant properties are well documented, most studies focus only on its identification and quantification, mainly through high-performance liquid chromatography (HPLC) for authenticity testing. Its specific contribution to the total antioxidant activity of MH remains underexplored. Consequently, no highly correlated biomarker has been established for assessing MH’s antioxidant effects. This highlights the need for the development of efficient and accurate methods to evaluate the antioxidant capacity of MH.

The antioxidant activity of MH was typically assessed using several common chemical assays, including 2,2′-azino-bis-3-ethylbenzothiazoline-6-sulfonic acid (ABTS), 2,2-diphenyl-1-picrylhydrazyl (DPPH), and ferric ion-reducing antioxidant power (FRAP). In addition, cellular antioxidant activity (CAA) assays were widely used to assess the antioxidant potential of food products [16]. Stagos et al. [17] investigated the antioxidant activity of various honey types using DPPH and ABTS methods, revealing that antioxidant activity differed across testing methods and concentrations. Roongpet et al. [18] compared the antioxidant activity of MH with Thai honey using the FRAP method, finding that certain Thai honey exhibited higher antioxidant properties than MH. The ABTS and DPPH methods are effective for assessing the overall antioxidant potential of honey, reflecting the overall activity of its antioxidant components, including polyphenols, flavonoids, and phenolic acids [19]. In contrast, the FRAP method mainly reflects the reducing power of specific antioxidants, such as vitamins and minerals, and has a more narrow detection range [20]. The combination of ABTS and DPPH assays offered a more complete antioxidant evaluation, with DPPH targeting a single free radical, while ABTS covers a broader spectrum. However, these chemical assays did not reflect the bioavailability and cellular uptake of antioxidants, which were essential for understanding their real-world efficacy and how effectively they acted within living organisms [21]. Consequently, cell-based assays like the CAA assay have gained recognition for their biological relevance in evaluating antioxidant activity in natural products [22,23].

The CAA assay, which uses the 2′,7′-Dichlorofluorescin diacetate (DCFH-DA) fluorescent probe to measure intracellular ROS levels, provides a more physiologically relevant assessment of antioxidant function. For instance, Portokalakis et al. [24] assessed the antioxidant activity of MH with different UMF grades in MCF-7 cells, finding a strong correlation between its cytotoxicity and total phenolic content. Deng et al. [23] compared the antioxidant activity of buckwheat honey and MH using HepG2 cells and found that buckwheat honey, despite having lower MGO levels, exhibited stronger antioxidant activity than MH. However, the complex matrix of honey, rich in sugars and polyphenolic compounds, can interfere with the results of such assays, leading to potential inaccuracies [25]. To date, most studies have focused on comparing the antioxidant activity of MH with that of other types of honey using the aforementioned methods. However, no study has systematically compared different antioxidant evaluation methods for MH itself, nor assessed their applicability, stability, or reliability. As a result, a standardized system for evaluating the antioxidant activity of MH has not yet been established.

This study aims to compare and evaluate the antioxidant capacities of MH using both chemical antioxidant assays and the CAA assay. Additionally, we validated the reproducibility and stability of the CAA method to assess its reliability in antioxidant research. The findings will provide a more accurate and reliable basis for evaluating the antioxidant potential of MH and its active compounds.

## 2. Materials and Methods

### 2.1. Material and Reagents

2,2-diphenyl-1-picrylhydrazyl (DPPH), 2,2′-azino-bis-3-ethylbenzothiazoline-6-sulfonic acid (ABTS), 2′,7′-dichlorofluorescin diacetate (DCFH-DA) probe, 2,2′-Azobis-2-amidinopropane dihydrochloride (ABAP), 3-Morpholinosydnonimine hydrochloride (SIN-1), and quercetin were purchased from Sigma-Aldrich, St. Louis, MO, USA. Methyl syringate (MSY) was purchased from TCI, Shanghai, China. Dulbecco’s modified eagle medium (DMEM), Phosphate-Buffered Saline (PBS), Penicillin–Streptomycin, and EDTA digestion solution were obtained from Sangon Biotech Co., Ltd., Shanghai, China. Fetal bovine serum (FBS) was purchased from Gibco, Thermo Fisher Scientific, Waltham, MA, USA. Acetonitrile (HPLC grade) used for high-performance liquid chromatography (HPLC) analysis was purchased from Thermo Fisher Scientific, Waltham, MA, USA. HepG2 and Caco-2 cell lines were provided by Suzhou Cos9x Biotech Co., Ltd., Suzhou, China. Manuka honey with UMF grades 5+, 10+, 15+, and 20+ was provided by the Technic Center of Zhejiang Entry–Exit Inspection and Quarantine Bureau, Hangzhou, China.

### 2.2. Antioxidant Assay of DPPH

The 2,2-diphenyl-1-picrylhydrazyl (DPPH) was used to evaluate the antioxidant capacity of honey samples, following the method proposed by Cheng et al. [26], with appropriate modifications. A fresh DPPH working solution (0.2 nmol/L) was prepared using anhydrous ethanol. The stock honey solution was prepared by dissolving 25 g of Manuka honey (MH) in 225 mL of deionized water and further diluted to obtain honey working solutions at different concentrations (5, 10, 15, 20, 25, and 30 mg/mL). These solutions were prepared at 25 °C and 37 °C for comparison.

For the DPPH assay, 1 mL of each honey sample was mixed with 1 mL of the DPPH working solution. The reaction mixtures were incubated in the dark at 25 °C and 37 °C for 48 h. Absorbance readings were measured at 517 nm using a microplate reader at multiple time points (0.5, 1, 3, 6, 24, and 48 h), with 200 μL samples collected at each time.

A regression equation was plotted based on the experimental concentrations of honey samples and their corresponding scavenging rates. The concentration of the honey sample at a 50% scavenging rate, represented by IC_50_, was then calculated to allow direct comparison between different samples. The DPPH scavenging activity was calculated using the following equation:(1)$$A_{DPPH}=\left(1-\left(\frac{A_{1}-A_{2}}{A_{0}}\right)\right)\times 100\%$$
where A_1_ refers to the absorbance of the experimental group (1 mL sample + 1 mL DPPH solution); A_2_ refers to the absorbance of the control group (1 mL sample + 1 mL ethanol); and A_0_ refers to the absorbance of the blank group (1 mL ddH_2_O + 1 mL DPPH solution).

### 2.3. Antioxidant Assay of ABTS

The 2,2′-azino-bis-3-ethylbenzothiazoline-6-sulfonic acid diammonium salt (ABTS) assay was used to evaluate the antioxidant honey samples, following the method described by Garcia et al. [27]. The ABTS stock solution was prepared by dissolving 0.0045 g of ABTS (MW = 270.32 g/mol) in 1.1025 mL of distilled water. A potassium persulfate (K_2_S_2_O_8_) solution was prepared by dissolving 0.0025 g of K_2_S_2_O_8_ in 3.575 mL of distilled water. The 7.4 mM ABTS stock solution and the 2.26 mM K_2_S_2_O_8_ solution were mixed and left to react in the dark at 25 °C for 12–16 h before use.

The ABTS working solution was then prepared by diluting the stock solution with deionized water to achieve an absorbance of 0.7 ± 0.02 at 734 nm. For the ABTS assay, 3 mL of the ABTS working solution was mixed with 0.5 mL of the honey samples at various concentrations and incubated in the dark at 25 °C. Absorbance was measured at 734 nm using a microplate reader at multiple time points (0.5, 1, 3, 6, 12, 24, and 48 h).

To create a dose–response curve showing the relationship between the concentration of honey sample solution and the scavenging rate of ABTS radicals, the concentration of honey samples was calculated when the scavenging rate was 50%, with the result expressed as IC_50_. The ABTS scavenging activity was calculated by the following equation:(2)$$A_{ABTS}=\left(\frac{A_{0}-A_{1}}{A_{0}}\right)\times 100\%$$
where A_0_ refers to the absorbance of the control group (3 mL ABTS working solution) and A_1_ refers to the absorbance of the test group (0.5 mL honey + 3 mL ABTS working solution).

### 2.4. Cell Culture

HepG2 cells, a human hepatocellular carcinoma cell line, and Caco-2 cells, derived from a human colon adenocarcinoma, were both purchased from Starfish Biology (cell biotechnology company in Suzhou) and were cultured using the method of Wolfe and Liu [28]. Briefly, cells were grown in Dulbecco’s modified eagle medium (DMEM) at 37 °C with 5% CO_2_, supplemented with 10% fetal bovine serum, 100 U/mL of penicillin, and 10 mg/mL of streptomycin.

### 2.5. Assay of Cell Cytotoxic

Methyl syringate (MSY) and quercetin were dissolved in DMSO and diluted with serum-free culture medium, ensuring a final DMSO concentration of <0.2%. The cytotoxicity of MH and MSY was carried out following the protocols previously described by Wofle et al. [29] with some modifications. HepG2 cells were seeded in 96-well microplates at a density of 2 × 10^4^ cells per well in 100 μL of culture medium. Caco-2 cells were seeded at 3 × 10^4^ cells per well under the same conditions. Both cell types were incubated at 37 °C for 24 h.

After incubation, the medium was removed, and the cells were washed with PBS. Fresh medium containing various concentrations of MH (5, 10, 15, 20, 25, and 30 mg/mL) or MSY (0.25, 0.5, 1, and 2 mg/mL) was then added. The cells were incubated again at 37 °C for another 24 h. Control groups were treated with serum-free culture medium only. After treatment, all cells were washed with PBS and supplied with fresh medium containing 10% CCK-8 solution. They were incubated for 1–2 h at 37 °C.

Finally, absorbance was measured at 450 nm using a microplate reader. Samples that caused a significant decrease in absorbance compared to the control group (*p* < 0.05) were considered cytotoxic. Cell viability (V%) was calculated using the following formula:(3)$$V\%=\frac{A_{T}-A_{B}}{A_{S}-A_{B}}\times 100\%$$
where A_T_ refers to the absorbance of the control group cells (no honey, no serum); A_s_ refers to the absorbance of the treated group (with honey or MSY); and A_B_ refers to the absorbance of the blank (no cells).

### 2.6. Lipid Peroxidation Levels

HepG2 cells were seeded in 6-well plates at a density of 2 × 10⁵ cells/mL and cultured for 24 h. Caco-2 cells were seeded at 3 × 10⁵ cells/mL in 2 mL of standard culture medium per well and incubated under the same conditions. Cells were divided into three groups: blank, control, and Manuka honey (MH) treatment groups. The MH groups received MH at concentrations of 5, 10, 15, and 20 mg/mL, dissolved in serum-free medium. The blank and control groups were treated with serum-free medium only.

After 24 h of pre-treatment, the medium was removed. The control and MH groups were then exposed to 1.5 mmol/L of 3-Morpholinosydnonimine hydrochloride (SIN-1) for 24 h to induce oxidative damage. The blank group continued to be cultured in serum-free medium without SIN-1. After incubation, all cells were washed twice with PBS.

Next, 150 μL of cell lysis buffer was added to each well, and cells were lysed for 2 min. The lysates were transferred to centrifuge tubes and centrifuged at 12,000 r/min for 10 min. The supernatants were collected, and intracellular malondialdehyde (MDA) levels were measured using an MDA assay kit, following the manufacturer’s instructions. SIN-1 was dissolved in distilled water to prepare a 10 mM stock solution and stored at −20 °C.

### 2.7. Assay of Cellular Antioxidant Activity (CAA)

The CAA assay was based on the method previously described by Wolfe et al. [28], with some adjustments. HepG2 and Caco-2 cells were seeded at the previously described densities in 96-well black, flat-bottom plates, using 100 μL of medium per well. Cells were incubated at 37 °C for 24 h. After reaching confluence, the medium was removed, and cells were washed with PBS. Next, 100 μL of medium containing the test samples (MH, MSY, and quercetin) and 25 μM 2′,7′-Dichlorofluorescin diacetate (DCFH-DA) was added to each well (in triplicate).

Cells were incubated for 1 h, then washed twice with 100 μL of PBS. Then, 600 μmol/L of 2,2′-Azobis-2-methylpropionamidine dihydrochloride (ABAP) dissolved in dissolved in Hank’s Balanced Salt Solution (HBSS) was added to the cells. The plate was placed in a fluorescence microplate reader. Fluorescence was recorded every 5 min for 1 h, using excitation at 485 nm and emission at 538 nm. DCFH-DA was dissolved in DMSO at 50 mM and stored at −20 °C. The ABAP stock solution (60 mM) was freshly prepared weekly and diluted to 60 μM with Hank’s solution before use. All stock solutions were stored at −20 °C and used within one month to ensure experimental reproducibility.

After cellular uptake, DCFH-DA was hydrolyzed by intracellular esterases to non-fluorescent DCFH, which was then oxidized by reactive oxygen species (ROS) to form the fluorescent compound dichlorofluorescein (DCF). The fluorescence intensity of DCF (excitation: 485 nm, emission: 538 nm) was measured to evaluate the intracellular oxidative status. The CAA was calculated by the following equation:(4)$$CAA\;unit=100-{(AUC}_{S}{/AUC}_{C})\times 100$$
where AUC_S_ was the integrated area under the fluorescence time curve of the sample, and AUC_C_ was the area under the fluorescence time curve of the control.

### 2.8. Determination of MSY in MH

The concentration of methyl syringate (MSY) in Manuka honey (MH) was quantified using high-performance liquid chromatography (HPLC), following a modified method described by Yoji Kato et al. [30]. Briefly, MH samples were dissolved in a solution of water and 10% acetonitrile to a final concentration of 0.1 g/mL. Prior to analysis, the solutions were filtered through a 0.22 μm membrane. A standard MSY stock solution was prepared at 1 mg/mL and serially diluted with water to obtain a range of standard working solutions from 0.25 mg/L to 100 mg/L. HPLC analysis was performed using an Acquity UPLC BEH C18 column (2.1 mm × 50 mm, 1.7 μm). The mobile phase consisted of a gradient system of acetonitrile (solvent A) and water (solvent B), operating at a flow rate of 0.3 mL/min. MSY was detected at a wavelength of 276 nm.

The gradient elution program was as follows: 0–3 min: 10% acetonitrile (A); 3–5 min: linear increase to 30% A; and 5–10 min: linear decrease back to 10% A (starting from 4.1 min).

### 2.9. Statistical Analysis

All data are presented as mean ± standard deviation (SD). Comparisons between group means were assessed using ANOVA, followed by multiple comparisons with the *T*-test. The IC_50_ values for cytotoxicity of the honey samples were calculated using IBM SPSS Statistics 26. Statistical significance was defined as *p* < 0.05, with a 95% confidence level.

## 3. Results

### 3.1. Effect of Temperature and Reaction Time on Chemical Assays

As shown in Figure 1B, in the ABTS assay, MH at concentrations ranging from 5 to 30 mg/mL reached maximum scavenging capacity within the experimental timeframe. Notably, 30 mg/mL MH achieved equilibrium as early as 0.5 h. In contrast, during the DPPH assay, MH at the same concentrations displayed sustained antioxidant activity over time, remaining in the kinetic phase without reaching a clear plateau. The scavenging ability of MH increased with higher concentrations. Specifically, 30 mg/mL MH reached its maximum scavenging rate at 24 h, whereas lower concentrations exhibited a gradual increase in scavenging capacity throughout the duration of the assay. Compared to DPPH, the ABTS assay had a faster response time, achieving equilibrium within 30 min, while DPPH required longer incubation for stabilization.

To evaluate the influence of temperature on free radical scavenging, the DPPH assay was performed at 37 °C (body temperature) and compared to 25 °C (room temperature). As shown in Figure 1A and Figure 2A, the DPPH scavenging activity of MH (5–30 mg/mL) increased over time at both temperatures. However, equilibrium was reached earlier at 37 °C, indicating enhanced scavenging kinetics under physiological conditions. In Figure 2B, the IC_50_ values significantly decreased over time at both 25 °C and 37 °C (*p* < 0.01), with consistently lower IC_50_ values at 37 °C. The values at 37 °C stabilized after 1 h, showing a significant difference only at the 0.5 h mark (*p* < 0.001), while those at 25 °C did not stabilize until 24 h, with significant differences across all earlier time points (*p* < 0.05).

### 3.2. Effect of Representative Compounds on Cellular Methods

The cytotoxicity of MH was assessed in HepG2 and Caco-2 cells at concentrations ranging from 5 to 30 mg/mL. As shown in Figure 3, treatment with 25 and 30 mg/mL MH significantly reduced cell viability to below 80% (*p* < 0.001). Specifically, HepG2 cell viability dropped to 70.23 ± 7.07% and 59.33 ± 8.97%, while Caco-2 cell viability decreased to 65.84 ± 9.6% and 67.93 ± 7.64%, respectively. Conversely, concentrations between 5 and 20 mg/mL of MH maintained cell viability above 85% in both cell lines. Therefore, 20 mg/mL was selected as the maximum safe concentration for subsequent cellular assays.

To induce oxidative stress in cells, the optimal concentration of SIN-1 was determined based on achieving 50–60% cell viability, inducing oxidative damage without causing complete cell death. Based on this criterion, the selected concentrations were 1.75 mmol/L for HepG2 and 1.5 mmol/L for Caco-2, as shown in Figure 4. These concentrations were chosen for subsequent experiments to ensure consistent oxidative stress induction across both cell lines.

The intracellular fluorescence intensity varied with DCFH-DA probe concentration. Higher concentrations resulted in faster stabilization and even signal saturation, while lower concentrations showed a slower or negligible increase. As shown in Figure 5, in HepG2 cells, 10 μM DCFH-DA exhibited a continuous increase over 60 min without stabilization. Similarly, 20 μM DCFH-DA in the Caco-2 cells displayed the same trend, indicating that probe concentration influences signal dynamics.

To evaluate the protective effect of MH, intracellular malondialdehyde (MDA) levels were measured after SIN-1-induced oxidative stress. As shown in Figure 6, lipid peroxidation levels significantly increased to 136.63 ± 19.46% in HepG2 (*p* < 0.01) and 155.42 ± 6.93% in Caco-2 (*p* < 0.001). Pre-treatment with 15 mg/mL and 20 mg/mL MH significantly reduced lipid peroxidation to below baseline levels, showing strong antioxidant protection. In HepG2 cells, levels decreased to 77.79 ± 6.33% and 53.46 ± 14.06%, respectively (*p* < 0.001). In Caco-2 cells, lipid peroxidation was reduced to 81.21 ± 8.45% and 72.12 ± 9.31% (*p* < 0.001). Lower concentrations of MH (5 mg/mL and 10 mg/mL) exhibited variable effects. In HepG2 cells, lipid peroxidation levels were 117.40 ± 7.93% (*p* < 0.05) and 89.90 ± 5.33% (*p* < 0.001), respectively. In Caco-2 cells, levels at 5 mg/mL and 10 mg/mL, respectively, were 115.86 ± 5.91% and 111.91 ± 11.92%, showing no significant difference compared to the oxidative damage model.

The SIN-1-induced experiment showed a nonlinear relationship between MH concentration and lipid peroxidation. As shown in Figure 7, the CAA assay revealed a significant positive correlation between MH concentration (5–20 mg/mL) and CAA fluorescence units. The correlation coefficients were R^2^ = 0.97 for HepG2 and R^2^ = 0.92 for Caco-2. In contrast to the lipid peroxidation assay, which required a 24 h incubation, the CAA assay allowed immediate detection of antioxidant activity, making it more time-efficient and concentration-sensitive for evaluating intracellular antioxidant capacity.

### 3.3. Effect of Sugar-Reduced Honey on CAA

The quercetin equivalent antioxidant content was quantified using both chemical assays and cell-based methods to compare the antioxidant capacities of MH and MSY. The results consistently demonstrated that MSY exhibited significantly higher antioxidant capacity across all methods. In the DPPH assay, MSY showed an antioxidant capacity of 75.45 ± 0.92 mg quercetin/100 g, which was approximately four times greater than that of MH (17.44 ± 3.17 mg quercetin/100 g). In the ABTS assay, MSY reached 91.76 ± 0.66 mg quercetin/100 g, nearly three times higher than (27.08 ± 3.21 mg quercetin/100 g). When evaluated in Caco-2 and HepG2 cells, MSY’s antioxidant capacity remained markedly higher. In Caco-2 cells, MSY showed an antioxidant capacity of 97.35 ± 8.13 mg quercetin/100 g, nearly 15 times greater than that of MH (6.24 ± 3.57 mg quercetin/100 g). In HepG2 cells, MSY recorded an antioxidant value of 255.06 ± 10.53 mg quercetin/100 g, approximately 23 times higher than that of MH (11.09 ± 4.08 mg quercetin/100 g).

The high sugar content in honey may contribute to osmotic pressure, which can interfere with cell membrane transport and affect intracellular processes. To address this, a sugar-removal treatment was performed on MH. As shown in Figure 8, the sugar-removed honey exhibited significantly enhanced antioxidant activity in the CAA assay compared to untreated honey (*p* < 0.001). Furthermore, the quercetin equivalent values of sugar-removed honey were significantly higher in both HepG2 and Caco-2 cells (*p* < 0.05). Notably, there was no significant difference between results obtained in HepG2 cells and those measured using the ABTS assay, which is known for its speed and sensitivity. Given the advantages in time efficiency and the ability to reflect in vivo-like activity, the CAA method is recommended for intracellular antioxidant detection.

### 3.4. Methodology Assessment

As shown in Table 1, the intra-day precision (intra-RSD) of the CAA method in HepG2 cells ranged from 3.16% to 4.83%, while the inter-day precision (inter-RSD) ranged from 6.26% to 7.51%. Similarly, in Caco-2 cells, the intra-RSD ranged from 4.98% to 6.01%, and the inter-RSD ranged from 8.90% to 9.52%. These results confirm that, under optimized conditions, the CAA method demonstrates high reproducibility and stability, with intra-RSD values below 5% and inter-RSD values below 10% in HepG2 cells. This indicates that the CAA assay is suitable for evaluating antioxidant capacity in cytological studies. Details on the specific values are provided in Appendix A.

### 3.5. Effect of Single Components on Antioxidant Activity Assessment

The MSY content in UMF 10+ MH was quantified using HPLC, with the calibration curve equation: y = 0.8204x + 0.337, with R^2^ = 0.999. The measured MSY content in UMF 10+ MH was 106.30 ± 6.50 mg/kg. Based on the quercetin equivalent calculations in Paragraph 3.3, MSY contributed approximately 0.080 mg quercetin/100 g MH in the DPPH assay and 0.0097 mg quercetin/100 g MH in the ABTS assay. In cell-based assays, its contributions were 0.027 mg quercetin/100 g MH in HepG2 cells and 0.01 mg quercetin/100 g MH in Caco-2 cells. These correspond to only 0.04% (DPPH), 0.03% (ABTS), 0.24% (HepG2), and 0.17% (Caco-2) of the total antioxidant activity of MH. These results indicate that MSY contributes no more than 0.5% to the overall antioxidant activity of Manuka honey in both chemical and cellular models. Although MSY showed strong antioxidant activity based on quercetin equivalents, its total content in MH was relatively low. Thus, its individual contribution to the overall antioxidant activity of MH was limited, suggesting that the antioxidant effects of MH are likely the result of synergistic interactions among multiple polyphenolic compounds.

### 3.6. Effect of Different UMF Grades on Antioxidant Activity in the CAA Assay

To evaluate whether the CAA method effectively detects the antioxidant activity of MH, different UMF-grade honeys were tested. As shown in Figure 9, honey concentrations ranging from 5 to 20 mg/mL showed a high coefficient of determination (R^2^ > 0.90) in both cell models, indicating good detection performance. Notably, the linear relationship was more consistent and reliable in the HepG2 cells (R^2^ > 0.96) compared to the Caco-2 cells (R^2^ > 0.92). The optimized CAA method demonstrated good detection capability for different MH samples.

## 4. Discussion

To better evaluate the antioxidant activity of Manuka honey (MH), we compared its performance at 25 °C and 37 °C. By optimizing in vitro assays with a reaction temperature of 37 °C and carefully controlling the reaction time, we aimed to better mimic physiological conditions. The ABTS method exhibited faster and more efficient radical scavenging across all MH concentrations compared to the DPPH assay. This is attributed to the higher reactivity of the ABTS radical cation, which enables faster electron transfer reactions, whereas the DPPH radical, with its stable unpaired electron, required slower hydrogen donation for neutralization [31,32]. Notably, the IC_50_ values decreased at both 25 °C and 37 °C, with consistently lower values observed at 37 °C. At 37 °C, the scavenging reaction stabilized within 1 h, whereas at 25 °C, equilibrium was only attained after 24 h. This suggests that MH exhibits significantly enhanced antioxidant activity at 37 °C. This enhanced efficiency may result from interactions between methyl syringate (MSY) and methylglyoxal (MGO), which likely stabilize MSY and inhibit degradation, thereby enhancing its antioxidant function [33]. Additionally, higher temperatures may promote the Maillard reaction, facilitating the conversion of dihydroxyacetone (DHA) into MGO, which would, in turn, improve the stability of phenolic compounds and strengthen free radical scavenging capacity [30,34].

MSY consistently exhibited higher antioxidant activity than MH at all concentrations, with low-concentration MSY outperforming high-concentration MH. The antioxidant activity of MSY is primarily attributed to its phenolic hydroxyl (-OH) and methoxy (-OCH_3_) groups, which enhance its ROS scavenging ability [35]. In addition to direct free radical neutralization, MSY can activate AMP-activated protein kinase (AMPK), a key oxidative stress regulator, which upregulates endogenous antioxidant enzymes such as superoxide dismutase (SOD) and catalase (CAT) [36,37]. Kirkpatrick et al. [35] identified MSY as the main contributor to the antioxidant activity of MH polyphenols, further supporting its strong bioactivity. The MSY content in UMF 10+ MH was quantified as 106.30 ± 6.50 mg/kg, consistent with previous reports ranging from 25.86 to 144.37 mg/kg [38]. While MSY exhibited stronger antioxidant activity than MH in both chemical and cellular assays, its overall concentration to MH’s antioxidant capacity was limited due to its low concentration. This emphasized the importance of evaluating the full spectrum of phenolic and flavonoid compounds in MH, as their synergistic interactions a interactions significantly impact antioxidant efficacy [39]. However, the synergistic effect between MSY and other MH phenolics was found to be minimal. Thus, MSY alone could not achieve maximal antioxidant potential within the complex honey matrix, highlighting the need to assess MH as a whole system in antioxidant studies [13].

Different cell types, such as HepG2 and Caco-2, assess the antioxidant capacity and explore the use of antioxidant components in the body [40]. Sugar-reduced MH exhibited significantly higher antioxidant activity, especially in the HepG2 cell experiment, where its antioxidant capacity was nearly double that of untreated MH. This suggested that removing sugars may increase the bioavailability or activation of antioxidant compounds. The enhanced activity may be attributed to the concentrated phenolic content in sugar-reduced honey, allowing for broader interactions and synergistic enhancement of antioxidant effects [13]. Interestingly, there was no significant difference in antioxidant activity between sugar-reduced MH in HepG2 cells and regular honey of the ABTS assay, indicating that methodological optimization can yield comparable antioxidant results across systems. MSY showed higher antioxidant capacity in HepG2 cells than in Caco-2 cells at the tested concentrations, potentially due to the hepatoprotective properties of syringic acid and its derivatives, including MSY [30]. In contrast, the difference in antioxidant activity between sugar-reduced and regular honey was less pronounced in Caco-2 cells. This may be similar to the characteristics of MSY, which has specific receptors on liver cell membranes, enhancing its antioxidant response. Overall, sugar-reduced honey showed a significant increase in antioxidant activity across different cell lines. Although the degree of enhancement varied, the antioxidant effect was consistently superior, especially in HepG2 cells, where it exhibited greater efficiency than in Caco-2 cells.

## 5. Conclusions

This study systematically optimized and compared chemical and cell-based methods for evaluating the antioxidant activity of Manuka honey (MH). Results showed that among the chemical assays, the ABTS method exhibited higher detection efficiency and responsiveness than the DPPH method, and increasing the reaction temperature improved the detection efficiency of the DPPH assay. Cellular antioxidant evaluation identified ≤ 20 mg/mL as the optimal testing concentration for MH, and sugar-reduced honey displayed a more accurate representation of its antioxidant potential. Additionally, the optimized cellular antioxidant activity (CAA) assay demonstrated strong linearity and reproducibility in assessing MH’s antioxidant activity. Its advantages in detection efficiency, biological relevance, and precision suggest its potential as a reference method for evaluating the antioxidant capacity of honey and other functional food products. Furthermore, although methyl syringate (MSY) exhibited significantly higher antioxidant activity than MH in both chemical and cellular assays, its low abundance contributed no more than 0.5% to the overall antioxidant capacity of MH. Therefore, the evaluation of MH’s antioxidant potential should focus on its overall antioxidant capacity rather than individual compounds. Overall, this study provides a validated methodological framework, supporting the CAA assay as a promising standardized approach for intracellular antioxidant assessment in Manuka honey research.

## Figures and Tables

**Figure 1 foods-14-01341-f001:**
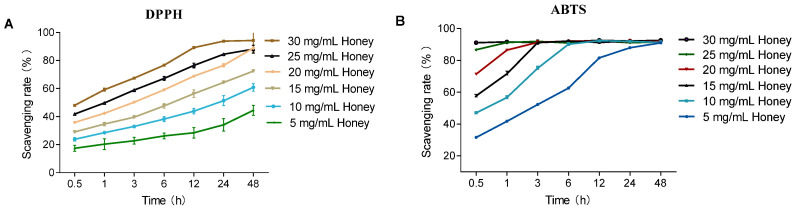
Effect of reaction time on the DPPH and ABTS radical scavenging capacities of MH. Note: (**A**) shows that the DPPH scavenging activity of MH (5–30 mg/mL) over time (0.5–48 h) at 25 °C; (**B**) shows that the ABTS scavenging activity of MH (5–30 mg/mL) over time at 25 °C.

**Figure 2 foods-14-01341-f002:**
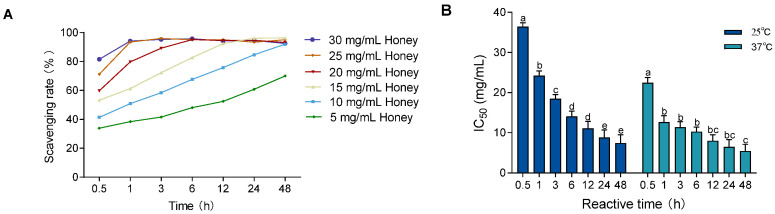
Effect of temperature and reaction time on DPPH radical scavenging ability of MH. Note: (**A**) shows the DPPH scavenging rate at 37 °C; (**B**) shows the change in IC_50_ values over time at both 25 °C and 37 °C. Data are presented as mean ± standard deviation from three independent experiments. Different letters (a, b, c, d, e) indicate significant differences among different reaction time groups (*p* < 0.05). The letter "bc" indicates no significant difference between groups b and c (*p* > 0.05).

**Figure 3 foods-14-01341-f003:**
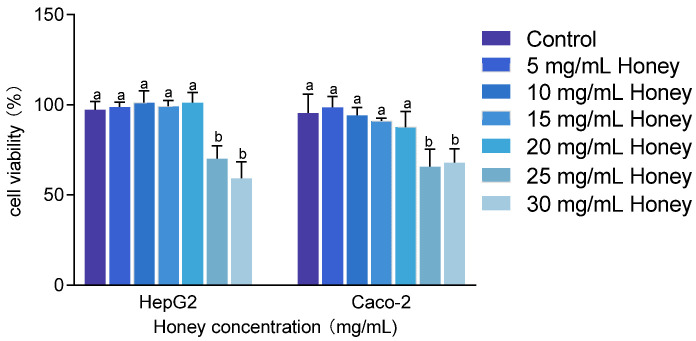
Cytotoxic effects of honey on HepG2 and Caco-2 cells. Note: Data are presented as mean ± standard deviation from three independent experiments. Using the untreated group as the 100% baseline, different letters (a, b) indicate statistically significant differences (*p* < 0.05).

**Figure 4 foods-14-01341-f004:**
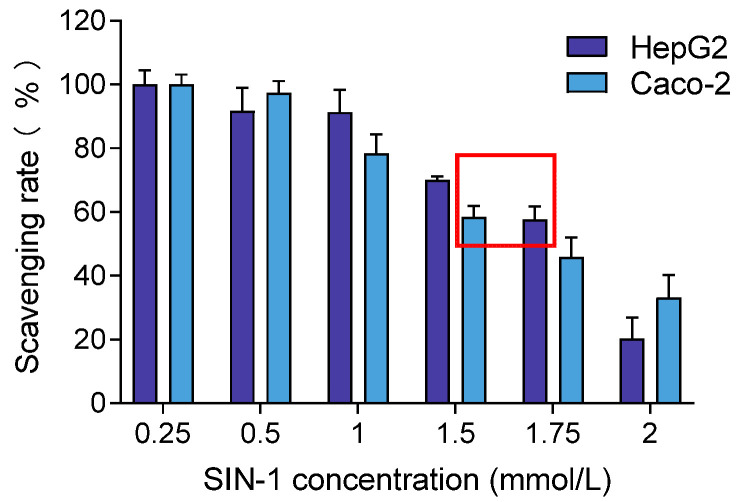
Effects of different concentrations of SIN-1 on HepG2 and Caco-2 cell viability. Note: The red box means the concentration of SIN-1 producing 50–60% cell viability in HepG2 and Caco-2.

**Figure 5 foods-14-01341-f005:**
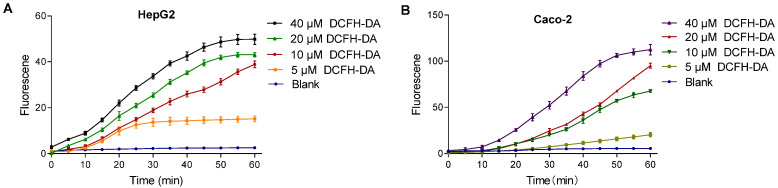
Effect of DCFH-DA concentrations on fluorescence in HepG2 (**A**) and Caco-2 (**B**) cells. Note: (**A**) dictates the fluorescence intensity changes of the DCFH-DA probe (5–40 μM) in HepG2 cells; (**B**) dictates the fluorescence intensity changes of the DCFH-DA probe (5–40 μM) in Caco-2 cells.

**Figure 6 foods-14-01341-f006:**
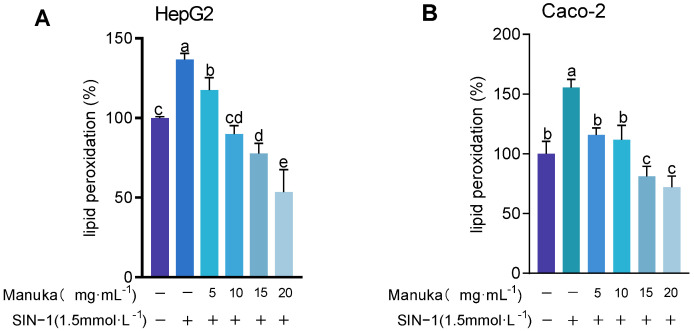
Lipid peroxidation results of MH in HepG2 (**A**) and Caco-2 (**B**) cells under SIN-1 induced damage. Note: Lipid peroxidation values were expressed as MDA content, presented as mean ± standard deviation. Using the untreated group as the 100% baseline, different letters (a, b, c, d, e) indicate statistically significant differences (*p* < 0.05).

**Figure 7 foods-14-01341-f007:**
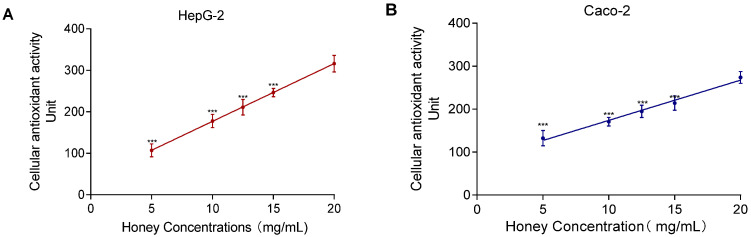
MH antioxidant activity and dose–response effects in HepG2 (**A**) and Caco-2 (**B**). Note: Data represent mean ± standard deviation from three independent experiments. Significance of differences between all concentrations and the highest concentration was determined using the *t*-test, with *** indicating *p* < 0.001.

**Figure 8 foods-14-01341-f008:**
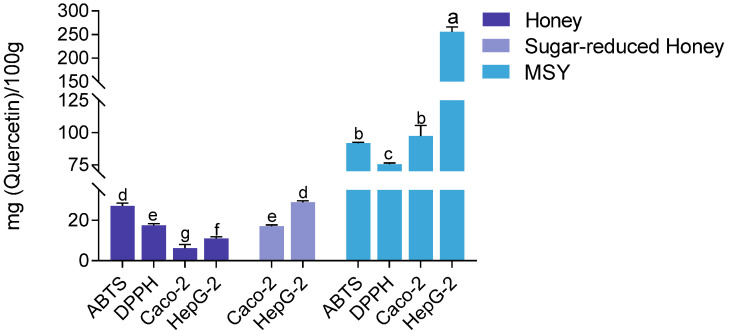
Comparison of antioxidant capacities of MSY, sugar-reduced honey, and untreated MH across different assays. Note: Statistical differences were evaluated using the *t*-test. Different letters (a, b, c, d, e, f, g) indicate significance at *p* < 0.05, *p* < 0.01, and *p* < 0.001, respectively. Discontinuities (\\) indicated axis breaks to resolve the vast difference in activity values between samples.

**Figure 9 foods-14-01341-f009:**
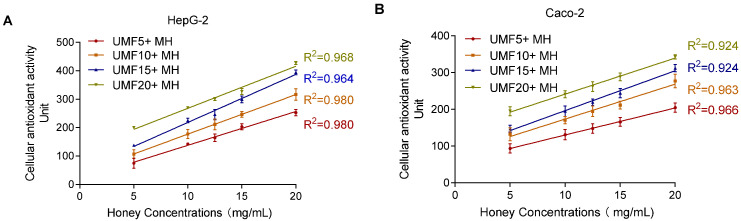
Effect of different concentrations (5–20 mg/mL) of MH measured by CAA assay on HepG2 (**A**) and Caco-2 (**B**).

**Table 1 foods-14-01341-t001:** Precision of the CAA method (inter-day and intra-day) for different UMF 10+ MH samples in HepG2 and Caco-2 cells (*n* = 3).

Honey Sample	HepG2		Caco-2	
	Intra-RSD (%)	Inter-RSD (%)	Intra-RSD (%)	Inter-RSD (%)
A	4.83	7.10	5.61	9.52
B	4.57	7.51	6.01	9.23
C	3.16	6.26	4.98	8.90

Note: Honey samples A, B, and C represent different batches of UMF 10+ MH. Three intra-day and three inter-day repetitions were conducted, each involving six sample groups per batch, to ensure data consistency and reliability.

## Data Availability

The data supporting this study’s findings are available from the author, Yafen Zhang, upon reasonable request.

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
