# Peer review of "Optimizing the Methodology for Antioxidant Activity Analysis of Manuka Honey"

_foods, 2025, doi:10.3390/foods14081341_

Round 1
Reviewer 1 Report
Comments and Suggestions for Authors
Dear authors, the work is valuable, it required thought and a lot of work, but in my opinion the topic should be more precise. Additionally, there are many editorial and punctuation errors in the work, which should be corrected with special attention. There is no summary of conclusions in the work, which should be supplemented. This gives a full picture of the work and a lesson for the readers. Additional notes below:
- Very numerous editorial errors, e.g. no space before citations. The brackets with citations should be directly after the cited name. In many places there are no dots at the end of the sentence, in general the entire paper should be corrected carefully taking into account the rules of punctuation.
- L 26 Leptospermum scoparium is a Latin name and should be written in italics
- L32 unnecessary space after the bracket
- L32-L36 too long sentence, split into 2
- L50 phenolics - should be plural
- L52 and L54 polyphenols - should be plural
- L57-L59 should be written as polyphenols from fruits as a functional additive to honey because it is unclear in the current version. In addition, it cannot be generalized that the correlation between polyphenols and antioxidant activity is weak in honeys because very often it is strong.
- L59 no period at the end of the sentence
- L62 (-OCH3) should be
- L76 remove the period before the bracket
- L76 Stagos et al. [17] should be
- L78 citation as above and in the whole text the same
- L82-L84 DPPH and ABTS methods do not allow for the detection of individual compounds (polyenols, flavonoids, etc.) in honey, but for the assessment of the antioxidant potential of their mixture present in honey (i.e. the entire product)
- L93 DCFH-DA expand the abbreviation
- L95 citation
- L95 UMF expand the abbreviation
- L97 citation
- L115 expand the abbreviations
- L116 expand MSY
- L117 and L119 are they the same manufacturer described differently
- L126 citation – cite the name of the author of the method
- L127 why ethanol? Methanol is usually used
- L130 punctuation (commas)
- L132 "of six concentrations to a tube" is not needed here
- L135 were not are
- L136-L139 sentence incomprehensible
- L142 period at the end
- L147 complete description of preparation of reagents (what solvent?)
- L167 MSY was a control? How prepared?
- L177 CCK small letters or capitals?
- L199 what is SIN-1?
- L208 Corning? Why capital?
- L212 ABAP expand abbreviation
- L238 and many more {Error}
- L378 UMF10 what is that?
- Paragraph 3.5 was the concentration of MSY itself adjusted to the real content in MH?
- The paper lacks a summary or conclusions.
- Figure 8 what do the gaps in the MSY bars mean?
- All enlarged graphs are difficult to read
- Reference: Check the correctness of the source literature, also taking into account punctuation
Comments on the Quality of English Language
The submitted work is valuable in terms of content, its execution required a lot of work and thought, additional aspects were taken into account, such as the high sugar content in honey. However, in my opinion, the topic should be more precisely specified, taking into account the assessment of antioxidant activity using cell cultures. In addition, there are many editorial and punctuation errors in the work, which should be corrected with special attention. The same applies to the language, it should be corrected. There is no summary in the work, which should be supplemented to give a full picture of the work.
Reviewer 2 Report
Comments and Suggestions for Authors
General comments
The evaluation of antioxidant activity of honey requires the application of different methods that allow determining the bioactive potential of the different compounds present in honey which act by different mechanisms of action.
Obtaining a methodology that allows standardizing the determination of the antioxidant activity of honey, in general, would be very useful and interesting. However, from my point of view this work does not cover a this aim and is insufficient and not applicable to the generality when testing a single sample of honey and two chemical tests and a CCA assay with two cell lines.
In my opinion the study should include different varieties of honey or, in the case of using only MH, test samples with different grades of UMF. In this way it would be possible to evaluate the methods respond to the variations in the composition of honey, more specifically with different contents of antioxidant compounds.
Specific comments
Please, in the footnote of the pages indicate the correct name of the journal.
Line 26. Indicate the name of the specie Leptospermum scoparium in italics.
Lines 108-109 This statement does not pose an hypothesis and seems to create a bias towards a specific result.
Line 131 The expression “room temperature” is not accurate, I think that it is better the authors indicate only the temperature of the assay.
Line 167 I suggest …”was carried out following the protocols previously described by…” indicating the authors.
Line 206 I suggest …”was based pm the method previously described by…” indicating the authors.
Results section: please correct all the “Error! Reference source not found” that appear.
Line 257: In Figure 2C is not possible to verify the radical scavenging improvement at 37 ºC because all the results presented in this figure are at 37ºC.
Line 295: I suggest “Fluorescent probes at different concentrations of DCFH-DA significantly…”
Line 323: I suggest “…relationship between the concentration of SIN-1 and lipid peroxidation…”
Comments on the Quality of English Language
In my opinion the Quality of English Language must be improved.
Adverbs or adverbial expressions frequently appear incorrectly placed
There are paragraphs which are difficult to understand for example:
2.5. Assay of cell cytotoxic (especially in lines 167-177)
2.7. Assay of cellular antioxidant activity (CAA)
2.8. Determination of MSY in MH. A paragraph cannot begin in the way it appears in the manuscript.
Reviewer 3 Report
Comments and Suggestions for Authors
- The number of honey samples, when and the exact location from which they were taken have not been defined. Increasing the sample size would enhance the reliability of the results.
- How can you be certain it's Manuka honey? Have you performed a pollen analysis to verify it?
- In the manuscript is not mentioned which compounds fluoresce in the measured samples.
- How many repetitions were performed for the results presented in Tables 1 and 2?
Round 2
Reviewer 1 Report
Comments and Suggestions for Authors
Good job!
Reviewer 2 Report
Comments and Suggestions for Authors
The quality of the manuscript has been considerably improved. Thank you.
I would like to point out a little mistake: the footer on the first page needs to be corrected (it still shows the name of another journal).